# Towards the Sustainable Development Goal of Zero Hunger: What Role Do Institutions Play?

**Jalini Kaushalya Galabada**

Graduate School of International Relations, International University of Japan, 777 Kokusai-cho, Minami Uonuma-shi 949-7277, Niigata-ken, Japan; jalini1983@iuj.ac.jp

**Abstract:** Empirical research has aimed to substantiate the institution–food security nexus. However, institutional literature has largely overlooked the relationship between institutions and the sustainable development goal of zero hunger (SDG2). SDG2 is a multidimensional goal that extends beyond food security and requires comprehensive investigation. Therefore, this study explored the role of institutions in promoting SDG2 achievement using a panel dataset spanning 108 countries from 2000 to 2019. The institutional impact was evaluated using worldwide governance indicators, and the International Country Risk Guide (ICRG)'s political risk ratings. Simultaneous equation modeling was used as the estimation technique. According to the results, institutions showed a positive and highly significant association with SDG2 performance. All the dimensions of good governance promoted SDG2 performance. Except for maintaining law and order, all other dimensions of political risk indicators were found to improve SDG2 performance. This study also discovered significant evidence that voice and accountability, as well as the settlement and the prevention of conflicts, had the most substantial influences on SDG2 achievement. In developing countries, improving both the quality of governance and political stability had a comparatively higher impact on SDG2 performance than in developed countries. Furthermore, institutions showed a significant mediating impact on SDG2 performance via agricultural productivity and economic growth. Based on these findings, this study concluded that the pursuit of good governance and inclusive institutions could be instrumental in achieving SDG2.

**Keywords:** institutions; sustainable development goals; zero hunger; 2SLS; 3SLS

## 1. Introduction

The United Nations' Sustainable Development Goals (SDGs) have dominated global policy discourse and actions since 2015. SDG2, which is widely recognized as "zero hunger," pledges to "end hunger, achieve food security and improved nutrition and promote sustainable agriculture" [1]. This multidimensional goal includes several distinct targets that can be divided into three dependent parts: eliminating hunger and improving nutrition (social dimension), achieving food security by improving productivity and increasing income (economic dimension), and encouraging sustainable agriculture (environmental dimension) [2].

Even before the COVID-19 pandemic, the world had fallen off track in its efforts to end hunger in all its forms by 2030 [3]. Human-driven conflicts, climatic variations and extremes, economic downturns, and desert locust outbreaks were identified as primary threats to its progress [4–6]. Then, the devastating effects of the SARS-CoV-2 outbreak drastically escalated global hunger in 2020–2021, exacerbating the challenge of achieving SDG2. Furthermore, the novel coronavirus remains widespread at the date of this publication, and it is hard to predict when it will cease to be a threat. Therefore, accelerated and swift counterbalancing actions are needed to guarantee food security for all [7]. Each country's institutional framework has played a fundamental role in recovery during the global pandemic. Mollier et al. [2] emphasized that coordinated and cohesive policies, in

collaboration with appropriate institutions, contribute to the development gains during complex circumstances by minimizing adverse effects.

According to Zhou and Wan [8], the conventional arguments regarding resource endowments, the size of the country and the population, the status of economic development, and cultural, and social differences cannot fully explain the disparities in food security across countries. Paarlberg [9] argued that the inadequate performance of nation-states continues to be crucial in combating hunger while emphasizing the necessity of providing fundamental policies for the public good, such as civil peace, the rule of law, and investment in research and development, to ensure access to sufficient food. Echarren [10] stated that eliminating hunger required long-term, sustained political, economic, and social interventions supported by robust institutional frameworks. In addition, Zhou and Wan [8] emphasized that the differences in institutions between countries have been assumed to account for the differences between nations regarding food security status. The impact of institutions and governance on food (in)security and hunger was well researched in the literature, and its significance was empirically substantiated [11–15].

Meanwhile, several scholars [16–18] and the SDG framework itself (e.g., Goal 16) have highlighted the importance of governance in achieving SDGs. For example, SDG16 requires the installation of accountable, efficient, and inclusive institutions [1] (p. 25). Furthermore, Miyazawa and Zusman [19] suggested that poor institutions are a primary reason for the limited achievement of the millennium development goals (MDGs), the previous global effort to tackle developmental priorities.

Significant variation between nations is apparent upon comparing national SDG2 scores, which are based on a system developed by the Sustainable Development Solutions Network (SDSN), which monitors global and national progress toward SDGs. However, despite the variations in national SDG2 scores worldwide, sufficient research has not yet been conducted to examine how institutions and their diverse characteristics have impacted SDG2 performance using a cross-country empirical analysis. Substantial empirical evidence on the relationship between institutions and SDG2 could provide insights for the development of evidence-based policies to encourage national governments and international bodies to upgrade institutions, ultimately leading to the achievement of SDG2.

Therefore, this study employed a cross-country analysis investigating the nexus where institutions and SDG2 meet using a large longitudinal data set. The study explored the direct and mediating effects of institutions on the achievement of SDG2 using instrumental variables (IV) in the two-stage least squares (2SLS) and three-stage least squares (3SLS) estimation methods. The study used simultaneous equation models as a strategy to overcome unobserved heterogeneity and the potential endogeneity issues. The study relied on panel data for 108 countries representing all income categories across twenty years from 2000 to 2019. Among the various measurements used to capture the institutional impacts in previous empirical studies, worldwide governance indicators and political risk ratings issued by the International Country Risk Guide (ICRG) were used in this study.

To the best of the author's knowledge, this paper contributes to the existing body of knowledge by rigorously examining the institutional impact on achieving SDG2 for the first time. The author analyzed the effect of institutions not just on food security but also on SDG2 performance, which was comprehensively assessed by accounting for most of the dimensions addressed by SDG2. Furthermore, by applying the same six indicators used by the SDSN for tracking the progress of the diverse targets of SDG2, a composite SDG2 index was constructed to capture SDG2 performance in this study. Therefore, the dependent variable of the current analysis was significantly different compared to the dependent variables used in the previous relevant literature. Hence, this study expanded our understanding of the impacts on the sustainable development goal of ending hunger, which remains a primary challenge worldwide.

## 2. Literature Review

Academics and policymakers worldwide have increasingly realized that further progress of broad development goals may be plausible, but only if national institutions are strengthened. Empirical evidence has shown the significance of institutions in predicting a country's level of development [20,21]. The United Nations has emphasized that institutions and development are inextricably linked and mutually reinforcing. Institutional advancement at the national and international levels is critical for inclusive economic growth, sustainable development, combating poverty and hunger, and the holistic protection of human rights and fundamental freedoms.

### 2.1. Institutions

Various definitions for "institutions" can be found in the literature. North [22] defined institutions as "the rules of the game in a society or are the humanly devised constraints that shape human interactions" (p. 3); according to Hodgson [23], institutions are „systems of established and prevalent social rules that structure social interactions" (p. 2). A significant number of scholars have substantiated that institutions, along with technological developments, have been the principal determinants of long-term economic growth and economic development [21,22]. The variation between economic institutions, which is based on political influence, the reigning political system, and the type of political institutions, has been a fundamental influence on economic development [24]. Weingast [25] argued that political institutions influenced the extent to which economic markets were sustainable and the level of political risk that impacted economic actors. Therefore, attention to political institutions is crucial for an economic system to succeed. Inclusive political institutions have contributed to creating inclusive economic institutions by facilitating the equitable distribution of resources and private property rights and subsequently offered a level playing field for all stakeholders [26].

### 2.2. Relationship between Institutions and the "Zero Hunger" Goal

Though, to the best of the author's knowledge, hardly any cross-country empirical study has focused on the institution–SDG2 nexus, there have been ample conceptual and empirical studies that explored the institution–food (in)security nexus [8,9,11–15,27]. Vos [27] suggested that given the long-term global nature and public good of sustainable food and nutrition security, cohesive measures and immediate improvements in global food security governance would be required. Conversely, Paarlberg [9] argued that the challenge of hunger and food insecurity required an urgent focus on addressing national governance deficits. The author further stressed that some regions continued to suffer significant hunger due mainly to national, not global, governance deficits and failures. Therefore, the role of governance should not be compromised and must be comprehensively integrated when planning and implementing food and nutrition security approaches that respond to diverse and evolving needs by promoting priorities and activities across the entire government [28,29].

According to Uchendu and Abolarin [30], corruption has actively hindered the efforts of international and regional development organizations that work to resolve hunger and famine crises, as well as interrupting business operations. Similarly, Helal [31] stated that food security governance and overall governance have been among the key determinants of food insecurity and that corruption resulted when governance failures occurred. In addition, weak political institutions and poor governance have significantly contributed to corruption and the rapid decline in food security [30,31]. Zhou and Wan [8] suggested that the quality of political institutions was a significant factor in promoting food security. They also indicated that food safety might not be reinforced where such institutions were not functioning properly. For instance, the European Union has provided a legal framework for food safety and food security through specific legislation using regulations, directives and decisions to ensure the food security of the community [32–34].

The 2030 Agenda for Sustainable Development identified conflict as a critical obstacle to realizing food security, and action should be taken to substantially mitigate all kinds of violence, particularly terrorism and military conflicts [35]. Brinkman and Hendrix [36] stated that the emergence of food insecurity was due, in part, to the rise in food prices as the result of the increased risk of a democratic break-up, protests, civil war, and sectarian struggles. Political stability and institutional reforms are vital for ensuring a stable food supply, and, therefore, improved long-term food security has been affected by political and socioeconomic conditions. Sen [37] explained the consequences of conflicts on food security: a decrease in food production, food accessibility, welfare, and human capabilities through the collapse of health and healthcare facilities, the environment, civic infrastructure, and education.

Ogunniyi et al. [13] empirically tested the influence of governance quality on food and nutrition security using panel data from sub-Saharan African countries. They showed that government efficiency, political stability, democratic accountability, and the rule of law enhanced both nutrition and food security. To measure food and nutrition security, they used the average value of production and the average dietary energy supply adequacy, respectively, as proxies. In addition, a cross-country study conducted by Abdullah et al. [11] evaluated the effect of political risk and institutions on food security in 124 countries using dietary energy supply as a proxy. According to their findings, corruption, internal and external conflicts, military in politics, ethnic tensions, religious tensions, poor bureaucracy, and poor socioeconomic conditions negatively impact countries' food security. In contrast, law and order, government stability, investment profiles, and democratic accountability positively and significantly affect the food supply.

In summary, the current literature demonstrates the fundamental role of institutions in assuring food and nutrition security, which could subsequently end hunger. Therefore, based on scholarly investigations into the institution–food (in)security nexus, institutions may be a critical factor in the achievement of SDG2. Similarly, although normative and conceptual contexts between governance and SDGs have been widely acknowledged, the SDG2–institution nexus has not been an empirical focus, resulting in a significant gap in the author's understanding.

*2.3. Analytical Framework*

Based on extensive scholarly explanations and findings concerning the relationship between institutions and food security, an argument can be made for the direct and indirect (i.e., mediation) impact of institutions on the achievement of SDG2. The literature discussed in Section 2.2 explicitly elucidates the direct link between institutions and SDG2 performance.

Improving agricultural production efficiency and maximizing yields are crucial for food security. Pawlak and Kolodziejczak [38] explained that increased agricultural productivity through the adoption of novel farming technologies and improved extension and training facilities for farmers, along with an open-trade policy that would not harm domestic suppliers and consumers, could ensure food security. However, according to the International Food Policy Research Institute (IFPRI) [39], most measures to boost Africa's agricultural productivity have been hampered by political instability, conflicts, a lack of stable institutions, and ineffective policies. Scholars have stressed the significance of supportive policies and institutional initiatives to increase food security in countries confronted with the challenge of increasing agricultural productivity [38,40,41]. Therefore, an argument for the mediating effect of institutions on SDG2 performance through agricultural productivity can be established based on the established correlations between agricultural productivity and food security, as well as between institutions and agricultural productivity.

Throughout a country's development, food security, economic growth, and equitable income distribution interact via a mutually reinforcing mechanism [42]. According to Engel's law, the long-term solution to food security was to foster rapid economic growth that incorporated the poor. Dreze and Sen [43] coined the phrase "growth-mediated security" to describe this type of economic growth. According to a substantial body

of literature, economic prosperity bolsters food security [44–46]. Similarly, much of the research undertaken over many decades placed an overwhelming emphasis on the nexus of institutions and economic growth [24,47–50]. As a result, an argument can be made for the mediating effect of institutions on SDG2 performance via economic growth.

The three main pathways through which institutions might influence SDG2 performance are depicted in Figure 1: (1) they may have a direct impact on SDG2 performance; (2) institutions can influence agricultural productivity and, therefore, SDG2 performance; and (3) institutions can impact economic growth and thus the achievement of SDG2. Each of these channels portrayed in Figure 1 was individually explored in this research.

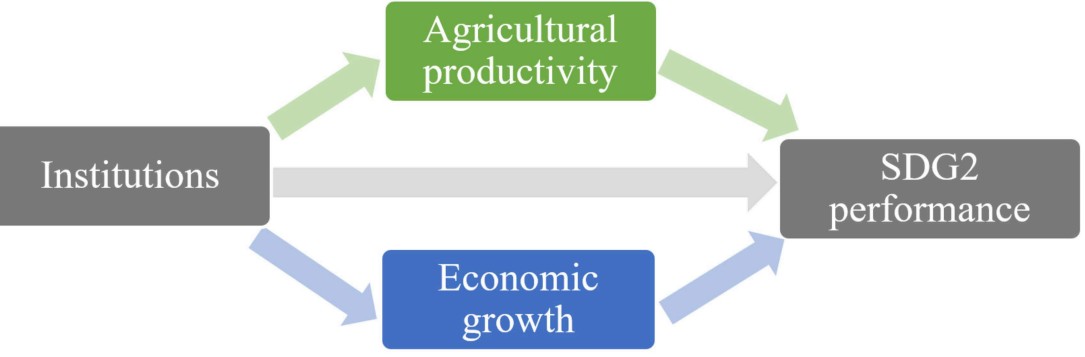

**Figure 1.** Path diagram of direct and indirect institutional impacts on the achievement of SDG2.

## 3. Methodology

### 3.1. Data

The study was performed using a panel dataset spanning 108 countries from 2000 to 2019. A composite index for SDG2 performance as a response variable was created using raw data acquired from the SDG index database developed by the SDSN. The indicators used to construct the SDG2 index were the prevalence of undernourishment in the population (%), the prevalence of stunting in children under five years of age (%), the prevalence of wasting in children under five years of age (%), the prevalence of obesity, i.e., body mass index (BMI) $\geq$ 30, in the adult population (%), the cereal yield (t/ha), and the sustainable nitrogen management index. In constructing the composite index, the same methodology performed to create the SDG index by the SDSN [51] was employed. First, the above indicators for SDG2 were normalized using the min–max method (Equations (1) and (2)). For the data series (i.e., the prevalence of undernourishment, prevalence of stunting in children under five years of age, prevalence of wasting in children under five years of age, and prevalence of obesity), a higher value implied an inferior performance level. In such cases, the estimates were rescaled using the min–max Equation (2) so that a higher value implied better performance [52]. Subsequently, the arithmetic mean of the values of the six indicators was used to determine the total goal score (SDG2 index). Finally, the SDG2 index was scaled from 0 to 1; higher values indicated better performance regarding SDG2.

$$I_x = \frac{x - x_{min}}{x_{max} - x_{min}}, \tag{1}$$

$$I_x = 1 - \frac{x - x_{min}}{x_{max} - x_{min}}, \tag{2}$$

where $I_x$, $x$, $x_{min}$, and $x_{max}$ denote a transformed indicator ranging from 0 to 1, the underlying raw data, the minimum value, and the maximum value, respectively.

The institutions, which were the independent variable of this study, were captured using two different measurement indices: worldwide governance indicators (WGIs) and political risk ratings issued by the ICRG [53]. First, a composite governance index (CGI) was constructed using the six WGIs, namely, voice and accountability (VA), political stability and absence of violence/terrorism (PS), government effectiveness (GE), regulatory quality

(RQ), the rule of law (RL), and control of corruption (CC) [54]. Principal component analysis (PCA) was performed to create the composite governance index using the cross-country data. Similarly, the political risk rating (PRR) consisted of twelve political risk components, namely, government stability (GVSTAB), socioeconomic conditions (SOECON), investment profile (INVPROF), internal conflict (INTCON), external conflict (EXTCON), corruption (CORRUP), military in politics (MILPOL), religious tensions (RELITEN), law and order (LAWORD), ethnic tensions (ETHTEN), democratic accountability (DEMACC), and bureaucracy quality (BUREAU). Each indicator was rated on a scale of either 0–12, 0–6, or 0–4, with higher scores suggesting lower political risk and stronger institutions. In this study, the latter two categories of indicators were rescaled to 0–12, ensuring unambiguous interpretation and ease of identifying the relative importance of each component [55]. Therefore, this study used a broader range of indicators for the quality of political and economic institutions to determine the comparative significance of those indicators on SDG2 performance.

As structural controls for examining the direct impact of institutions on SDG2, annual population growth (POPG), openness to trade (TO) measured by the sum of exports and imports as a share of GDP (%), education (EDU) measured by the number of years of schooling, and urbanization (URBN) measured by the proportion of urban dwellers as a percentage of the total population were used. In exploring the mediating effects of institutions on SDG2, the logarithm of agricultural value added per worker (lnAP) and the logarithm of GDP per capita (lnPG) were used to measure agricultural productivity and economic growth, respectively. In the 3SLS model estimation, the logarithm of the agricultural capital stock (lnAC), logarithm of agricultural land (lnAL), fertilizer consumption in kilograms per hectare of arable land (FERT), employment in agriculture as a percentage of total employment (AEMP), logarithm of gross fixed capital formation (lnCF), total natural resources rent as a percentage of GDP (NC), the sum of average years of secondary education completed and tertiary education completed among people over age 25 representing human capital (HC), and foreign direct investment net inflows as a percentage of GDP (FDI) were applied as control variables with the relevant mediator variable. The instrumental variables from previous studies on governance and sustainable development were used in this study [56]. In addition, as the instrumental variables, the classification of legal origin [57], the logarithm of settlers' mortality [21,58], ethnolinguistic divergence [56,59], latitude [57], the fraction of land within 100 km of the coast [60], a dummy variable for whether a country was landlocked, a financial openness index, and a terrain ruggedness index [61] were used.

### 3.2. Model Specification and Methods of Estimation

#### 3.2.1. Principal Component Analysis

The principal component analysis (PCA) was employed to develop CGI, which was one of the measurement indexes used in this study to capture institutional performance [54]. PCA applies an orthogonal transformation to transform the set of observations of likely correlated variables $(X_1, \dots, X_P)$ into a set of values of linearly uncorrelated variables $(PC_1, \dots, PC_K)$, which are referred to as principal components. The principal component variable CGI has the advantage of being orthogonal to the constructing variables while still capturing all common variability. The predicted model was depicted as follows:

$$CGI_{it} = (PC_{VA} \times VA_{it}) + (PC_{PS} \times PS_{it}) + (PC_{GE} \times GE_{it}) + (PC_{RQ} \times RQ_{it}) + (PC_{RL} \times RL_{it}) + (PC_{CC} \times CC_{it}), \quad (3)$$

where the subscripts i and t refer to the cross-section unit and period, respectively.

### 3.2.2. Simultaneous Equation Modeling

This study employed simultaneous equation modeling to investigate the direct and indirect effects of institutions on SDG2 performance.

Instrumental Variables (IV) and the Two-Stage Least Squares (2SLS) Method

The 2SLS regression analysis was applied with the chosen instrumental variables to evaluate the direct impact of institutions on SDG2 performance. The representations of the models adopted in this study were as follows:

$$SDG2_{it} = \alpha_1 CGI_{it} + \alpha_2 POPG_{it} + \alpha_3 EDU_{it} + \alpha_4 TO_{it} + \alpha_5 URBN_{it} + \varepsilon_{it}, \tag{4}$$

$$SDG2_{it} = \beta_1 PRR_{it} + \beta_2 POPG_{it} + \beta_3 EDU_{it} + \beta_4 TO_{it} + \beta_5 URBN_{it} + u_{it}, \tag{5}$$

$SDG2_{it}$ indicates the composite index for SDG2 performance. $CGI_{it}$ denotes the composite governance index, while $PRR_{it}$ denotes the political risk rating, which captures the effect of institutions. POPG, EDU, TO, and URBN are structural controls denoting population growth, education, trade openness, and urbanization, respectively. Variables $\alpha_1$–$\alpha_5$ and $\beta_1$–$\beta_5$ are the respective coefficients to be estimated, and $\varepsilon_{it}$ and $u_{it}$ represent the error terms. The use of instrumental variables in the IV–2SLS method used for estimating the above model addressed the potential endogeneity biases and furnished reliable estimates of the structural parameters.

A variety of diagnostic tests were employed to determine whether the model sufficiently described the relationships between the variables. The test for endogeneity detects the null hypothesis of whether the endogenous regressors used are indeed exogenous. The detection of significance ($p \leq 0.05$) rejected the null hypothesis of no endogeneity. The under-identification test is a Lagrange multiplier (LM) test that determines whether an equation is identified, i.e., whether the omitted instruments are correlated with the endogenous regressors. The LM version of the Anderson canonical correlation test verifies whether an equation is identified. The null hypothesis of weak instruments was tested using the Cragg–Donald F-test [62]. An instrument is considered "very strong" if the Cragg–Donald F-statistic is larger than 10% of the maximum IV size, as indicated by [62]. It is classified as "strong," "medium," and "weak" if the maximum IV sizes are between 10% and 15%, 15% and 20%, and 20% and 25% of the total, respectively. The validity test (overidentified restrictions test) determines whether any instrument is weak if associated with the error term. The Sargan test was used as a validity test in this analysis. The null hypothesis was that all instrumental factors in the stage 2 regression were uncorrelated with the error term, and the judgment of non-significance confirmed the null hypothesis.

Three-Stage Least Squares (3SLS) Method

The 3SLS method of simultaneous equation estimation explored the mediating effects of institutions on SDG2 performance while also capturing the direct effect. The 3SLS technique, proposed by [63], avoids endogeneity in the simultaneous equation system. It is an instrumental variable generalized least squares (IV-GLS) technique that ensures efficiency and consistency through appropriate weighting and instrumenting. The 3SLS method can compute all the coefficients of the overall system concurrently. Instrumental variables were critical in this scenario since certain control variables contained in the channel equation were endogenous in the system. Each equation in the system was considered to be at least identified. The coefficients of the parameters of interest explained the impact of a marginal variation in the independent variable. The coefficient of the independent variable in the channel equation was multiplied by the coefficient of the mediating variable in the central equation to show how the independent variable influenced the dependent variable through the mediating variable.

The simultaneous equation system was based on a cross-country SDG2 equation (Equation (6)) and two distinct equations for each channel via agricultural productivity and economic growth. The connection between institutions (INS) and SDG2 was simulated using lnAP and lnPG as mediating variables (MVs) (Equation (6)). In the model estimation, INS was captured via CGI and PRR.

$$SDG2_{it} = \beta_0^{SDG2} + \beta_1^{SDG2}INS_{it} + \beta_2^{SDG2}MV_{it} + \sum_{l=n}^{n} \beta_1^{SDG2}CV_{it}^{SDG2} + \varepsilon_{it}^{SDG2}, \quad (6)$$

where $CV_{it}^{SDG2}$ denotes the control variables. Equations (7) and (8) are cross-country channel equations in which the dependent variables were lnAP and lnPG, respectively, while the independent variable was INS.

$$lnAP_{it} = \beta_0^{lnAP} + \gamma_1^{lnAP}INS_{it} + \sum_{l=n}^{n} \beta_1^{lnAP}CV_{it}^{lnAP} + \varepsilon_{it}^{lnAP} \quad (7)$$

$$lnPG_{it} = \beta_0^{lnPG} + \gamma_1^{lnPG}INS_{it} + \sum_{l=n}^{n} \beta_1^{lnPG}CV_{it}^{lnPG} + \varepsilon_{it}^{lnPG} \quad (8)$$

The Sobel test was performed to determine the mediating effect [64]. The coefficients that characterized the effect of the channel variable, as well as the coefficients describing each channel variable's effect on the dependent variable, were of particular importance. The product of matching parameters on a particular channel path then provided the corresponding channel impact. When assessing the statistical significance of mediating effects, the standard error of $S_{ab}$ was calculated using the formula below:

$$S_{ab} = \sqrt{b^2 S_a^2 + a^2 S_b^2 + S_a^2 S_b^2} \quad (9)$$

where $S_{ab}$ is the standard error of $\gamma_1^m \beta_m^Y$. The variables a and b are $\gamma_1^m$ and $\beta_m^Y$, respectively; $S_a^2$ is the variance of the equation that explains the impact of INS on the channel variable; and $S_b^2$ is the variance of the equation narrating the impact of the channel variable on SDG2.

### 3.2.3. Pooled Ordinary Least Squares, Fixed Effects, and Random Effects Models

The pooled ordinary least squares (OLS), fixed effects (FE), and random effects (RE) models were performed to examine the robustness of the direct effect results. In addition, the Lagrange multiplier (LM) and the Hausman tests were used as diagnostic tests to evaluate whether the RE model was better than the pooled OLS estimation and to choose between the FE and RE models, respectively [65].

### 4. Results and Discussion

The author constructed a composite governance index using worldwide governance indicators. As a preliminary requirement for eligibility for the PCA, pairwise correlations (Table A1 in Appendix A) showed a significantly positive correlation between the indicators. According to the results (Table A2 in Appendix A), only the eigenvalue of component 1 (5.21954) was >1 and captured 87% of the internal system's variability.

Table 1 shows the summary statistics of the data used in the study. The SDG2 index ranged between 0.307 and 0.835. The sample's average country had met only 58 percent of the SDG2 goal. According to the minimum and maximum estimates, the composite governance index (CGI) varied between −5.206 and 4.751, while the political risk rating (PRR) ranged between 33.5 and 97. Figure 2 illustrates the geo–visualization of the SDG2 index, CGI, and PRR, which were the dependent and independent variables of the study.

**Table 1.** Summary statistics of the main variables.

| Variable | Number of Observations | Mean | Standard Deviation | Minimum | Maximum |
| --- | --- | --- | --- | --- | --- |
| SDG2 | 2160 | 0.58 | 0.083 | 0.307 | 0.835 |
| CGI | 2052 | 0 | 2.297 | −5.206 | 4.751 |
| PRR | 2160 | 66.866 | 12.468 | 33.5 | 97 |
| POPG | 2160 | 1.48 | 1.373 | −3.848 | 15.177 |
| TO | 1939 | 81.801 | 47.523 | 15.564 | 384.582 |
| EDU | 1518 | 61.243 | 39.891 | 1.121 | 163.935 |
| URBN | 2160 | 59.871 | 21.517 | 14.61 | 100 |
| lnAP | 2111 | 8.619 | 1.503 | 5.434 | 11.636 |
| lnAC | 2160 | 8.85 | 1.974 | 2.308 | 14.148 |
| lnAL | 2052 | 11.184 | 1.964 | 4.5 | 15.481 |
| FERT | 1980 | 158.37 | 213.251 | 0 | 2192.42 |
| AEMP | 2160 | 26.176 | 22.606 | 0.68 | 82.99 |
| lnPG | 2154 | 8.653 | 1.537 | 5.272 | 11.626 |
| lnCF | 1883 | 23.717 | 1.995 | 17.59 | 28.999 |
| NC | 2154 | 6.977 | 10.219 | 0 | 58.983 |
| HC | 1959 | 3.312 | 1.851 | 0.11 | 7.74 |
| FDI | 2152 | 5.79 | 20.461 | −58.323 | 449.083 |

Source: Author's estimation.

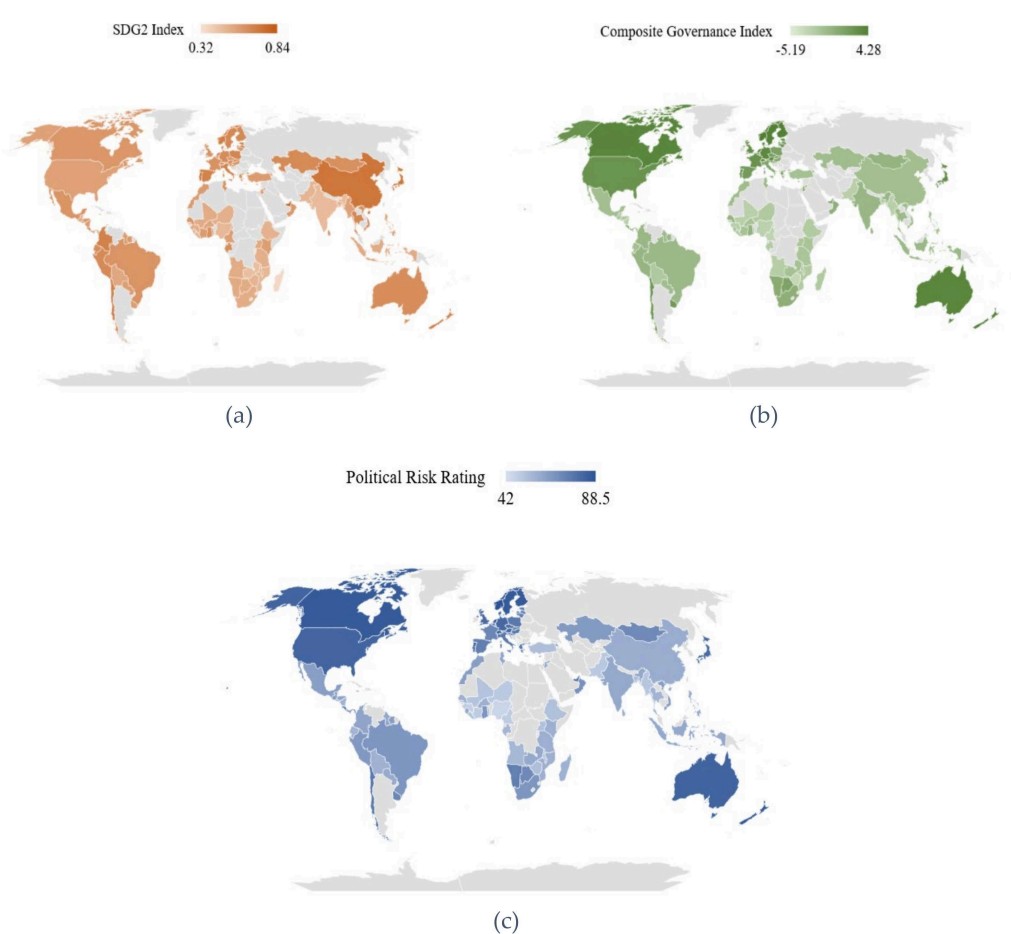

(a)

(b)

(c)

**Figure 2.** Geo-visualization of the dependent and independent variables across countries. (**a**) SDG2 index (dependent variable); (**b**) Composite Governance Index (independent variable); (**c**) Political Risk Rating (independent variable) Source: Author's creation.

As presented in Table 2, the correlations between SDG2 performance and both measurement indexes for institutions (CGI and PRR) were positive and significant.

**Table 2.** Correlation matrix.

| Variables | SDG2 | CGI | PRR | POPG | TO | EDU | URBN | lnAP | lnAC | lnAL | FERT | AEMP | lnPG | lnCF | NC | HC | FDI |
|---|---|---|---|---|---|---|---|---|---|---|---|---|---|---|---|---|---|
| SDG2 | 1.00 | | | | | | | | | | | | | | | | |
| CGI | 0.63 * | 1.00 | | | | | | | | | | | | | | | |
| PRR | 0.62 * | 0.93 * | 1.00 | | | | | | | | | | | | | | |
| POPG | −0.37 * | −0.41 * | −0.36 * | 1.00 | | | | | | | | | | | | | |
| TO | 0.28 * | 0.31 * | 0.35 * | −0.07 * | 1.00 | | | | | | | | | | | | |
| EDU | 0.33 * | 0.28 * | 0.30 * | −0.23 * | 0.09 * | 1.00 | | | | | | | | | | | |
| URBN | 0.59 * | 0.62 * | 0.60 * | −0.25 * | 0.26 * | 0.33 * | 1.00 | | | | | | | | | | |
| lnAP | 0.65 * | 0.82 * | 0.75 * | −0.40 * | 0.27 * | 0.33 * | 0.79 * | 1.00 | | | | | | | | | |
| lnAC | 0.18 * | 0.26 * | 0.26 * | −0.15 * | −0.14 * | 0.08 * | 0.20 * | 0.31 * | 1.00 | | | | | | | | |
| lnAL | −0.36 * | −0.25 * | −0.25 * | 0.04 | −0.57 * | −0.14 * | −0.26 * | −0.31 * | 0.36 * | 1.00 | | | | | | | |
| FERT | 0.40 * | 0.33 * | 0.31 * | −0.01 | 0.07 * | 0.24 * | 0.35 * | 0.36 * | 0.13 * | −0.21 * | 1.00 | | | | | | |
| AEMP | −0.66 * | −0.72 * | −0.69 * | 0.43 * | −0.31 * | −0.36 * | −0.82 * | −0.91 * | −0.26 * | 0.37 * | −0.37 * | 1.00 | | | | | |
| lnPG | 0.72 * | 0.84 * | 0.80 * | −0.42 * | 0.31 * | 0.35 * | 0.82 * | 0.92 * | 0.35 * | −0.28 * | 0.38 * | −0.90 * | 1.00 | | | | |
| lnCF | 0.44 * | 0.51 * | 0.44 * | −0.36 * | −0.17 * | 0.20 * | 0.45 * | 0.60 * | 0.62 * | 0.34 * | 0.24 * | −0.54 * | 0.66 * | 1.00 | | | |
| NC | −0.25 * | −0.41 * | −0.29 * | 0.45 * | −0.02 | −0.07 * | −0.08 * | −0.30 * | −0.07 * | 0.03 | −0.05 * | 0.23 * | −0.20 * | −0.22 * | 1.00 | | |
| HC | 0.56 * | 0.74 * | 0.68 * | −0.46 * | 0.26 * | 0.27 * | 0.67 * | 0.80 * | 0.25 * | −0.17 * | 0.21 * | −0.75 * | 0.82 * | 0.55 * | −0.21 * | 1.00 | |
| FDI | 0.09 * | 0.10 * | 0.12 * | −0.03 | 0.28 * | 0.05 | 0.10 * | 0.09 * | −0.03 | −0.24 * | −0.01 | −0.08 * | 0.07 * | −0.12 * | −0.02 | 0.07 * | 1.00 |

Source: Author's estimation. * $p < 0.1$

The uphill pattern shown in Figures 3 and 4 verified the positive correlations between the above variables. In addition, all the structural controls used in the analysis showed significant correlations. The below plots (Figures 3 and 4) suggested a linear association between the two variables in both cases.

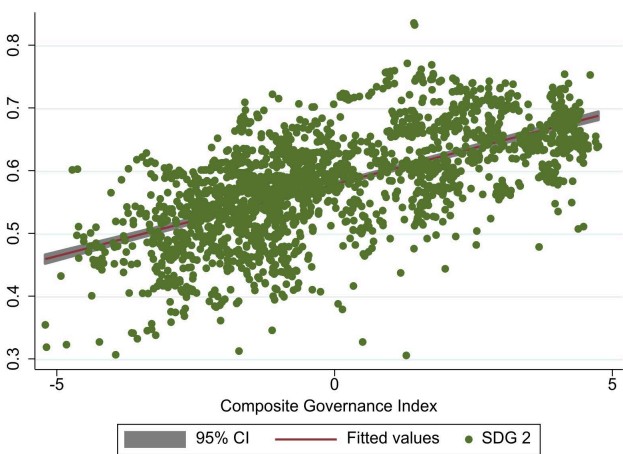

**Figure 3.** SDG2 performance against composite governance index. Source: Author's creation.

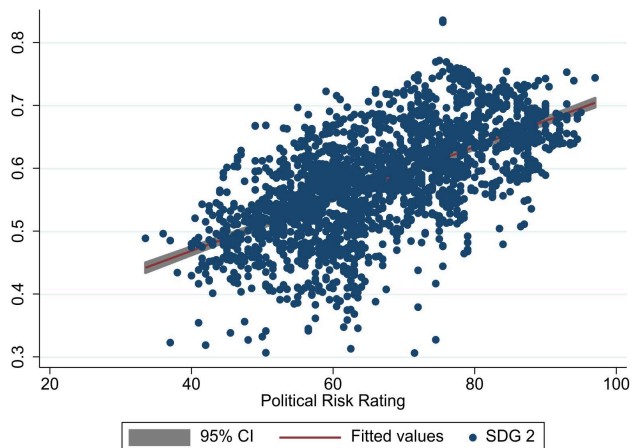

**Figure 4.** SDG2 performance against political risk rating. Source: Author's creation.

### 4.1. Direct Impact of Institutions on SDG2 Performance According to Worldwide Governance Indicators

Table A3 in Appendix A displays the results of the 2SLS estimation for governance against the SDG2 index and the impact of institutions on SDG2 performance. Model 1 was estimated using the composite governance index (CGI) as the independent variable to apprehend the overall effect on SDG2 performance. Individual governance indicators were used as independent variables in models 2–7 to assess the relative influence of diverse governance characteristics on achieving SDG2. The diagnostic analysis of the 2SLS regression suggested that the estimated models adhered to valid statistical criteria. The Anderson canonical correlation LM statistic rejected the null hypothesis that the equation was under-identified, which meant that the model was identified. As indicated by the results of the Cragg–Donald Wald F-statistic in Table A4, the instrumental variables applied in all the models were strong since the IV sizes were above 10% of the maximum IV size. According to the Sargan test results, the instrumental variables used in all the models were valid, indicating that the instrumental variables were uncorrelated with the errors, and there were no omitted variables in the model.

The main results depicted in Figure 5 illustrate the point estimates of the composite governance index (CGI) and all the discrete governance indicators. The results attest to

a highly significant positive relationship between governance and SDG2 performance, indicating that governance aspects played a decisive role in the progress toward achieving SDG2. Ceteris paribus, an increase in the composite governance index by one unit increased the SDG2 index by 0.0246 units on average.

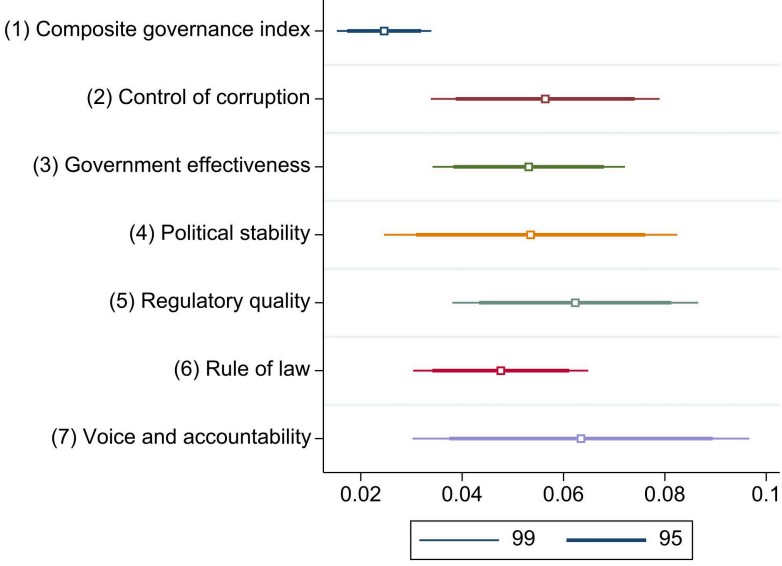

**Figure 5.** Impact of governance aspects on SDG2 performance. Dependent variable: SDG2 index. This figure was generated using coefplot in Stata [66]. It plots the coefficients from seven separate 2SLS estimations, each with the SDG2 index as the dependent variable and different independent variables. The horizontal lines extending from the circles represent 95% (thick lines) and 99% (thin lines) confidence intervals. All the regressions used the same control variables. Source: Author's creation.

Furthermore, voice and accountability (VA) showed the strongest influence among the individual governance indicators, while the rule of law (RL) showed the weakest influence on SDG2 performance. Voice and accountability enable society to take part in its choice of representatives and drive the government to address public concerns. Accountability has been crucial when addressing issues in food security initiatives and guaranteeing the efficient distribution of food supplies [67]. According to Asare-Nuamah [68], citizen participation in the government process through voice and accountability significantly improved Ghana's food security. Furthermore, Rocha Menocal and Sharma [69] emphasized that voice and accountability interventions impact global development goals. Although Glass and Newig [18] analyzed the effect of governance aspects on the performance of each SDG using a different set of variables to capture governance, the fitted model for SDG2 was non-significant. However, the finding of a positive association between governance and SDG2 performance in this study was in line with the explanations provided by the international development community, as well as with previous empirical results that examined the relationship between governance and food security [9,12,15,27].

*4.2. Direct Impact of Institutions on SDG2 Performance According to Political Risk Rating*

Figure 6 illustrates the coefficients of the 2SLS model estimation of the political risk rating (PRR) against the SDG2 index and shows the direct impact of institutions on SDG2 performance (Table A4 in Appendix A presents the complete results).

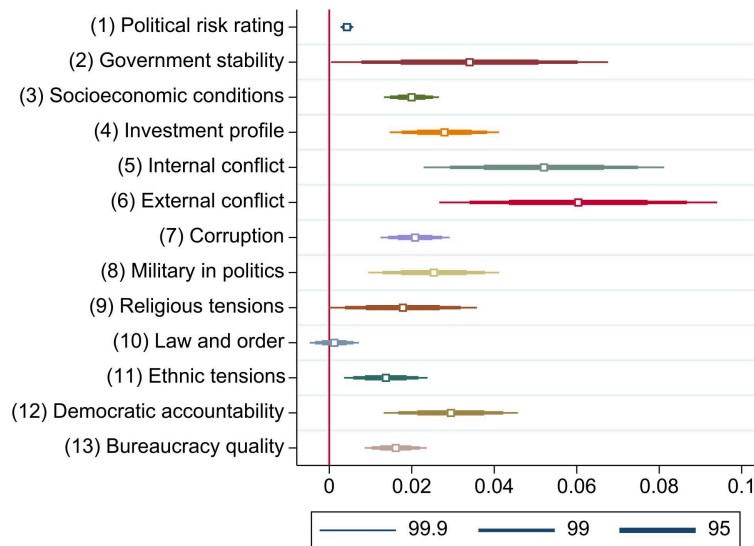

**Figure 6.** Impact of political risk aspects. Dependent variable: SDG2 index. Point estimates indicate the coefficients, while the thinnest to thickest horizontal lines indicate the 99.9% to 95% confidence intervals, respectively, using coefplot in Stata [66]. Source: Author's creation.

First, model 8 was estimated using the political risk rating (PRR) as the independent variable to encapsulate their overall impact on SDG2 performance. Then, models 9–20 were estimated using the individual risk components of the political risk rating as independent variables to examine the relative effect of diverse aspects of political risk on achieving SDG2. The composite political risk rating and all the individual risk components, except law and order, were found to have positive and significant relationships with SDG2 performance at $p < 0.01$, which indicated that political risk and political stability played a crucial role in the progress toward achieving SDG2 (Figure 6, Table A4). Ceteris paribus, an increase in the political risk rating by one unit increased the SDG2 index by 0.0043 units on average. The higher the risk point total, the lower the risk was, and the higher the SDG2 index, the higher the SDG2 performance was; the results indicated that a country's political stability provided a conducive environment for progress toward achieving SDG2. Furthermore, of the significant individual risk components at $p < 0.01$, external conflict (EXTCON) and internal conflict (INTCON) showed the most substantial influence, while ethnic tensions (ETHTEN) showed the weakest influence related to SDG2 performance. This appeared to corroborate the findings of previous empirical studies on the association between conflict and increased food insecurity [70–73].

According to the Food and Agriculture Organization (FAO) of the United Nations, estimates taken a few years before the COVID-19 pandemic suggested that the prevalence of undernourishment was higher in countries experiencing conflict, violence, and fragility than in countries that were not. Similarly, the progress toward reducing hunger and undernutrition had stalled or worsened in most conflict-affected countries. A national conflict can impact all four aspects of food security (i.e., availability, access, utilization, and stability) [74]. The accompanying instability of internal conflicts disrupted the delicate agricultural production cycle, destabilized markets and transportation networks delivering food to its final destination, and raised production costs [10]. Holleman et al. [74] highlighted that the impact of conflict on food and nutrition security was substantially worse when a conflict was prolonged and aggravated by a weak institutional response (fragility), along with other stress factors, such as droughts and other climatic uncertainties.

Recently, a study [11] investigated the impact of political risk and institutions on food security using robust econometric techniques and political risk rating (PRR) data as the measurement of political risk. However, the findings of that study were not directly comparable to this study, as dietary energy supply served as a proxy for food security

in [11], while this study employed a composite index using six indicators representing the SDG2 performance as the ultimate goal of ensuring food security.

### 4.3. Contrasting the Direct Impact of Institutions on SDG2 in Developing and Developed Countries

By partitioning the sample into two groups, namely, developed and developing countries, this study investigated whether the influence of institutions on SDG2 performance was impacted by a country's level of development. Table 3 displays the findings of the estimated 2SLS models for developed and developing nations individually.

**Table 3.** Direct impact of institutions and SDG2 performance: developing countries versus developed countries.

| Variables | CGI | | PRR | |
|---|---|---|---|---|
| | Developing [1] | Developed [1] | Developing [1] | Developed [1] |
| CGI | 0.0535 *** | 0.0201 *** | | |
| | (2.8208) | (5.6399) | | |
| PRR | | | 0.0106 *** | 0.0035 *** |
| | | | (4.0459) | (5.5532) |
| POPG | −0.0069 | 0.0065 ** | −0.0122 ** | 0.0026 |
| | (−1.4433) | (2.1635) | (−2.2748) | (1.0806) |
| TO | 0.0000 | −0.0001 | −0.0007 ** | −0.00003 |
| | (0.0284) | (−1.0895) | (−2.1888) | (−0.4331) |
| EDU | 0.0001 | 0.0001 | 0.0002 | 0.0001 |
| | (1.1214) | (1.5664) | (1.2623) | (1.1611) |
| URBN | 0.0009 *** | −0.0003 | −0.0004 | 0.0002 |
| | (4.3532) | (−1.2171) | (−0.9205) | (1.1974) |
| Constant | 0.5798 *** | 0.6146 *** | −0.0186 | 0.3407 *** |
| | (15.5771) | (31.9198) | (−0.1424) | (7.9969) |
| Observations | 495 | 524 | 364 | 559 |
| Anderson canon. | 11.923 | 72.633 | 17.672 | 72.894 |
| corr. LM statistic | (0.0026) | (0.000) | (0.000) | (0.000) |
| Cragg–Donald Wald F-statistic | 6.022 [b] | 41.597 [b] | 9.109 [b] | 41.388 [b] |
| Sargan statistic | 1.378 | 0.190 | 3.211 | 0.201 |
| | (0.240) | (0.663) | (0.073) | (0.654) |
| Endogeneity test | 8.030 | 8.776 | 44.921 | 10.277 |
| | (0.005) | (0.003) | (0.000) | (0.001) |
| Instruments | Seacoast | Latitude | Latitude | Latitude |
| | Landlocked | EF | EF | EF |

Note: All models were estimated using 2SLS estimation. Numbers within parentheses are the *t*-statistics of coefficients, except in the case of the Anderson canon. corr. LM statistic, Sargan chi-squared, and endogeneity test for which *p*-values are given. [b] Stock–Yogo weak ID test critical values at 10%, 15%, 20%, and 25% of the maximum IV sizes were 19.93, 11.59, 8.75, and 7.25, respectively. *** and ** indicate significance at the 1% and 5% levels, respectively. EF denotes ethnic–fractionalization. [1] In reference to the World Bank's country classification, high- and upper-middle-income countries were categorized as developed countries, while low- and lower-middle-income countries were categorized as developing countries for this analysis. Source: Author's estimation.

The results of the diagnostic tests confirmed that all models were substantially well specified. The institutional development, both in terms of governance and political risk, appeared to increase the potential for achieving SDG2 in both developing and developed countries. However, improving both the quality of governance and political stability had a higher positive impact on SDG2 performance in developing countries than in developed countries. The developed and developing countries were classified according to their per capita income, reflecting their economic development. Using the convergence theory, i.e., the "catch-up effect," introduced by Clark Kerr, the deviation of SDG2 performance can be explained. Fundamentally, the catch-up effect describes a phenomenon in which emerging countries expand more rapidly than wealthier countries. Furthermore, Góes [75] diagnosed the strikingly diverse dynamics of institutions in advanced and developing countries, which suggested that improving institutional quality yielded declining rewards.

Góes's emphasis on improved outcomes as a result of enhancing institutional quality in developing nations corroborates the current study's findings.

### 4.4. Indirect Impact of Institutions on SDG2 Performance

The coefficient estimates of the simultaneous equations, using agricultural productivity as the mediator variable, are shown in Table 4. Ceteris paribus, a 1% increase in agricultural productivity boosted SDG2 performance by 0.000127 units, while a one-unit increase in CGI increased agricultural productivity by 25.76% on average in the 3SLS model 21. According to model 22, a 1% increase in agricultural productivity improved SDG2 performance by 0.000141 units on average, while a one-unit rise in PRR improved agricultural productivity by 3.3%, ceteris paribus.

**Table 4.** Indirect impact of institutions on SDG2 performance through agricultural productivity.

| Variables | Model 21 | | Model 22 | |
|---|---|---|---|---|
| | **CGI as Independent Variable** | | **PRR as Independent Variable** | |
| | **SDG2** | **lnAP** | **SDG2** | **lnAP** |
| CGI | 0.0031 | 0.2576 *** | | |
| | (1.3378) | (15.6295) | | |
| PRR | | | 0.0004 | 0.0330 *** |
| | | | (1.0258) | (11.0458) |
| lnAP | 0.0127 ** | | 0.0141 *** | |
| | (2.5515) | | (3.3164) | |
| TO | 0.0005 *** | | 0.0005 *** | |
| | (6.0857) | | (5.9799) | |
| URBN | 0.0003 | | 0.0003 | |
| | (1.3102) | | (1.2907) | |
| EDU | 0.0002 ** | | 0.0002 ** | |
| | (2.2771) | | (2.1410) | |
| POPG | −0.0259 *** | | −0.0259 *** | |
| | (−7.7263) | | (−7.8664) | |
| lnAC | | 0.0317 ** | | 0.0287 ** |
| | | (2.4204) | | (2.0449) |
| lnAL | | 0.0649 *** | | 0.1000 *** |
| | | (3.6272) | | (5.2372) |
| FERT | | 0.0002 ** | | 0.0004 *** |
| | | (2.2903) | | (4.5090) |
| AEMP | | −0.0377 *** | | −0.0422 *** |
| | | (−23.0612) | | (−24.5324) |
| Constant | 0.4469 *** | 8.4401 *** | 0.4135 *** | 5.9000 *** |
| | (11.2857) | (42.0340) | (16.4842) | (19.1471) |
| Observations | 397 | 397 | 425 | 425 |
| R-squared | 0.5599 | 0.8872 | 0.5530 | 0.8598 |
| Sobel test | 0.0034 *** | | 0.0005 *** | |
| | (2.5167) | | (3.1796) | |

Note: All models were estimated using the 3SLS technique proposed by Zellner and Theil [63]. Numbers within parentheses are the *t*-statistics of coefficients. *** and ** indicate significance at the 1% and 5% levels, respectively. Source: Author's estimation.

When the Sobel test was performed to assess the impact of institutions on SDG2 via agricultural productivity, both for CGI and PRR, positive and significant results were found. This suggested that better agricultural productivity resulted from both strong governance and political stability, which improved SDG2 performance. When the CGI was strengthened by one unit, the results demonstrated that agricultural productivity had a 0.34% mediating effect on the SDG2 index. Comparably, when the PRR was increased by one unit, agricultural productivity had a 0.05% mediating effect on the SDG2 index. This agreed with the conceptualization of boosts in agricultural productivity as a significant mediating variable in the institution–food security nexus [38,40,41].

Table 5 shows the empirical results of the mathematical model with economic growth as the mediator variable. In the 3SLS model 23, ceteris paribus, a 1% gain in economic growth raised SDG2 performance by 0.000232 units on average, whereas a one-unit rise in CGI boosted economic growth by 33.24%. As reported regarding model 24, a 1% increase in economic growth increased SDG2 performance by 0.000225 units on average, while a one-unit increase in PRR improved economic growth by 4.85%. Furthermore, the direct model of the 3SLS estimation confirmed the robustness of the findings on the direct impact of both CGI and PRR on SDG2 performance using a 2SLS estimation.

**Table 5.** Indirect impact of institutions on SDG2 performance through economic growth.

| Variables | Model 23 | | Model 24 | |
|---|---|---|---|---|
| | CGI as Independent Variable | | PRR as Independent Variable | |
| | SDG2 | lnPG | SDG2 | lnPG |
| CGI | 0.0049 *** | 0.3324 *** | | |
| | (2.8267) | (26.0390) | | |
| PRR | | | 0.0010 *** | 0.0485 *** |
| | | | (3.8244) | (22.1286) |
| lnPG | 0.0232 *** | | 0.0225 *** | |
| | (5.4727) | | (6.2975) | |
| TO | 0.0002 *** | 0.0039 *** | 0.0001 *** | 0.0036 *** |
| | (3.1346) | (6.2912) | (3.0445) | (5.2787) |
| URBN | 0.0000 | | 0.0000 | |
| | (0.2568) | | (0.3346) | |
| EDU | 0.0001 ** | | 0.0001 ** | |
| | (2.3461) | | (2.2694) | |
| POPG | −0.0015 | | −0.0020 | |
| | (−0.8492) | | (−1.1686) | |
| lnCF | | 0.2427 *** | | 0.2545 *** |
| | | (19.6585) | | (19.8832) |
| NC | | 0.0166 *** | | 0.0053 * |
| | | (5.5545) | | (1.7653) |
| HC | | 0.1796 *** | | 0.2561 *** |
| | | (12.0645) | | (17.6937) |
| FDI | | 0.0006 | | 0.0014 |
| | | (0.5999) | | (1.2365) |
| Constant | 0.3708 *** | 1.9879 *** | 0.3119 *** | −1.6865 *** |
| | (11.6541) | (6.5738) | (19.6807) | (−5.6612) |
| Observations | 801 | 801 | 859 | 859 |
| R-squared | 0.5606 | 0.8744 | 0.5655 | 0.8515 |
| Sobel test | 0.0077 *** | | 0.0011 *** | |
| | (5.3532) | | (6.0592) | |

Note: All models were estimated using the 3SLS technique proposed by Zellner and Theil [63]. Numbers within parentheses are the *t*-statistics of coefficients. ***, **, and * indicate significance at the 1%, 5%, and 10% levels, respectively. Source: Author's estimation.

When evaluating the impact of institutions on SDG2 through economic growth using the Sobel test, both CGI and PRR showed significant positive results. This suggested that, through economic growth, both good governance and political stability improved SDG2 performance. In addition, the results showed that economic growth had a 0.77% mediating effect on the SDG2 index when the CGI increased by one unit. Similarly, in terms of PRR, economic growth had a 0.11% mediating effect on the SDG2 performance when the PRR increased by one unit. As a result, this finding authenticated the role of economic growth as a mediator in the link between institutions and food security found in previous research [42,43,76].

## 5. Robustness Check

A variety of estimating techniques, namely, pooled OLS, FE, and RE, were used to assess the robustness of the fitted models to account for the direct impact of institutions on SDG2 achievement (model 1 of Table A4 and model 8 of Table A5). The results of the robust models suggested a comparable institutional impact on SDG2 performance.

## 6. Conclusions

This study, first, revealed the likely impacts that institutions had on SDG2 achievement, expanding beyond food security, and hence, second, this research contributed to the empirical literature by examining the precise channels of influence on the overlooked institution–SDG2 nexus, such as direct and indirect effects. Rather than relying on a single measurement index to represent institutions, this research applied worldwide governance indicators and political risk ratings, both as a composite index and as individual indicators, to ascertain the overall institutional impact and the relative importance of each dimension regarding achieving SDG2. Using simultaneous equation modeling with longitudinal data to yield robust estimates was another distinctive feature of this study.

The findings on the first objective revealed that institutions had a positive and highly significant direct relationship with SDG2 performance. All aspects of good governance impacted SDG2 achievement. All other components of political risk indicators, except for maintaining law and order, improved SDG2 performance. This research also provided concrete evidence that enhanced voice and accountability, as well as the resolution and prevention of conflicts, had the most significant impact on SDG2 accomplishment. Improvements in governance quality and political stability significantly influenced SDG2 performance in developing countries compared to advanced nations. According to the results of the second objective, institutions had a mediating impact on SDG2 performance via agricultural productivity and economic growth.

In light of the above findings, this study suggested that the pursuit of good governance and inclusive institutions could be instrumental in policy decisions targeting the achievement of SDG2. As Rocha Menocal and Sharma [69] emphasized, the ability of citizens to express and exert their opinions has been vital to the eradication of poverty. Voice and accountability boost food security through monetary stability. Therefore, reinforcing voice and accountability by altering the institutional and power dynamics that impact citizens' behaviors and attitudes to foster social, political, and economic freedoms may provide a substantially favorable environment for the attainment of SDG2. Concurrently, this study suggested that restraining outbreaks of conflict, ensuring food security during the conflict, and facilitating post-conflict recovery through effective policy interventions may promote the achievement of SDG2. Furthermore, by focusing on the mediating role of institutions, effective, cohesive policymaking through an inclusive institutional framework may boost agricultural productivity and economic growth, which would be conducive to realizing the goal of zero hunger.

**Funding:** This research and the APC were funded by the Japan International Cooperation Agency (JICA).

**Institutional Review Board Statement:** Not applicable.

**Informed Consent Statement:** Not applicable.

**Data Availability Statement:** The data that support the findings of this study are available from the author upon reasonable request.

**Acknowledgments:** I would like to thank the anonymous reviewers for their insightful comments on the earlier version of this paper. I am also grateful to N.S. Cooray and K. Yamada from the International University of Japan for their guidance on this paper and the research behind. Furthermore, I wish to sincerely acknowledge the generous support given by my scholarship donor, Japan International Cooperation Agency (JICA), for my educational endeavor.

**Conflicts of Interest:** The author declares no potential conflict of interest.

## Appendix A

**Table A1.** Pairwise correlations for the worldwide governance indicators.

| Variables | CC | GE | PS | RQ | RL | VA |
|---|---|---|---|---|---|---|
| CC | 1.0000 | | | | | |
| GE | 0.9471 * | 1.0000 | | | | |
| PS | 0.7613 * | 0.7350 * | 1.0000 | | | |
| RQ | 0.9149 * | 0.9468 * | 0.7357 * | 1.0000 | | |
| RL | 0.9614 * | 0.9598 * | 0.7787 * | 0.9428 * | 1.0000 | |
| VA | 0.8416 * | 0.8318 * | 0.7070 * | 0.8599 * | 0.8538 * | 1.0000 |

* $p < 0.01$. Source: Author's estimation.

**Table A2.** Principal components (eigenvectors).

| Component | Eigenvalue | Difference | Proportion | Cumulative |
|---|---|---|---|---|
| Comp1 | 5.27448 | 4.91838 | 0.8791 | 0.8791 |
| Comp2 | 0.3561 | 0.142692 | 0.0594 | 0.9384 |
| Comp3 | 0.213408 | 0.129079 | 0.0356 | 0.9740 |
| Comp4 | 0.0843286 | 0.0449294 | 0.0141 | 0.9881 |
| Comp5 | 0.0393992 | 0.00711496 | 0.0066 | 0.9946 |
| Comp6 | 0.0322842 | | 0.0054 | 1.0000 |

| Variable | Comp1 | Comp2 | Comp3 | Comp4 | Comp5 | Comp6 |
|---|---|---|---|---|---|---|
| CC | 0.4213 | −0.1263 | −0.2366 | −0.6339 | −0.3619 | 0.4666 |
| GE | 0.4212 | −0.2245 | −0.3005 | 0.0958 | 0.8043 | 0.1606 |
| PS | 0.3626 | 0.9265 | −0.0198 | 0.0744 | 0.0429 | 0.0478 |
| RQ | 0.4194 | −0.2150 | −0.0735 | 0.7394 | −0.4339 | 0.1938 |
| RL | 0.4267 | −0.1085 | −0.2097 | −0.1572 | −0.1444 | −0.8465 |
| VA | 0.3945 | −0.1311 | 0.8967 | −0.1097 | 0.1057 | −0.0041 |

Source: Author's estimation.

**Table A3.** Direct impact of institutions on the performance of SDG2 according to worldwide governance indicators.

| | Model 1 | Model 2 | Model 3 | Model 4 | Model 5 | Model 6 | Model 7 |
|---|---|---|---|---|---|---|---|
| **Variables** | **CGI** | **CC** | **GE** | **PS** | **RQ** | **RL** | **VA** |
| CGI | 0.0246 *** | | | | | | |
| | (8.718) | | | | | | |
| CC | | 0.0564 *** | | | | | |
| | | (8.245) | | | | | |
| GE | | | 0.0531 *** | | | | |
| | | | (9.244) | | | | |
| PS | | | | 0.0535 *** | | | |
| | | | | (6.106) | | | |
| RQ | | | | | 0.0623 *** | | |
| | | | | | (8.482) | | |
| RL | | | | | | 0.0488 *** | |
| | | | | | | (9.107) | |
| VA | | | | | | | 0.0634 *** |
| | | | | | | | (6.303) |
| POPG | −0.0012 | −0.0052 ** | −0.0026 | −0.0038 | −0.0009 | −0.0055 *** | 0.0024 |
| | (−0.561) | (−2.522) | (−1.277) | (−1.465) | (−0.406) | (−2.981) | (0.741) |
| TO | −0.0000 | 0.0001 * | 0.0001 | −0.0002 ** | −0.0000 | 0.0001 | 0.0001 |
| | (−0.124) | (1.856) | (1.087) | (−2.003) | (−0.399) | (0.944) | (1.480) |
| EDU | 0.0002 *** | 0.0002 *** | 0.0001 *** | 0.0002 *** | 0.0002 *** | 0.0002 *** | 0.0002 *** |
| | (3.494) | (3.196) | (2.869) | (3.698) | (3.962) | (4.180) | (4.211) |
| URBN | 0.0004 ** | 0.0001 | 0.0004 ** | 0.0010 *** | 0.0004 ** | 0.0006 *** | 0.0007 *** |
| | (2.050) | (0.275) | (2.551) | (6.346) | (2.181) | (3.434) | (3.509) |

**Table A3.** *Cont.*

| | **Model 1** | **Model 2** | **Model 3** | **Model 4** | **Model 5** | **Model 6** | **Model 7** |
|---|---|---|---|---|---|---|---|
| **Variables** | **CGI** | **CC** | **GE** | **PS** | **RQ** | **RL** | **VA** |
| Constant | 0.5496 *** | 0.5619 *** | 0.5379 *** | 0.5362 *** | 0.5341 *** | 0.5372 *** | 0.5049 *** |
| | (44.159) | (39.474) | (49.594) | (36.293) | (46.475) | (49.019) | (46.075) |
| Observations | 860 | 860 | 860 | 873 | 860 | 860 | 873 |
| Anderson canon. | 123.385 | 99.351 | 154.349 | 79.578 | 119.602 | 152.770 | 71.687 |
| Corr. LM statistic | (0.000) | (0.000) | (0.000) | (0.000) | (0.000) | (0.000) | (0.000) |
| Cragg–Donald Wald F–statistic | 71.440 [b] | 55.706 [b] | 93.290 [b] | 43.429 [b] | 68.896 [b] | 92.129 [b] | 25.795 [a] |
| Sargan statistic | 0.308 | 1.854 | 0.149 | 1.607 | 0.642 | 0.016 | 3.951 |
| | (0.579) | (0.173) | (0.700) | (0.205) | (0.423) | (0.899) | (0.139) |
| Endogeneity test | 33.476 | 37.716 | 24.527 | 29.675 | 37.164 | 27.249 | 24.299 |
| | (0.000) | (0.000) | (0.000) | (0.000) | (0.000) | (0.000) | (0.000) |
| Instruments | Latitude | Latitude | Latitude | EF | Latitude | Latitude | EF |
| | EF | EF | EF | GEL | EF | EF | UKL |
| | | | | | | | GEL |

Note: All models were estimated using 2SLS estimation. Numbers within parentheses are the *t*-statistics of coefficients, except in the case of the Anderson canon. corr. LM statistic, Sargan chi-squared, and endogeneity test for which *p*-values are given. [a] Stock–Yogo weak ID test critical values at 10%, 15%, 20%, and 25% of the maximum IV sizes were 22.30, 12.83, 9.54, and 7.80, respectively. [b] Stock–Yogo weak ID test critical values at 10%, 15%, 20%, and 25% of the maximum IV sizes were 19.93, 11.59, 8.75, and 7.25, respectively. ***, **, and * indicate significance at the 1%, 5%, and 10% levels, respectively. EF, UKL and GEL denote ethnic–fractionalization, English legal origin and German legal origin, respectively. Source: Author's estimation.

**Table A4.** Direct impact of institutions on the performance of SDG2 according to the political risk rating.

| | Model 8 | Model 9 | Model 10 | Model 11 | Model 12 | Model 13 | Model 14 | Model 15 | Model 16 | Model 17 | Model 18 | Model 19 | Model 20 |
|---|---|---|---|---|---|---|---|---|---|---|---|---|---|
| **Variables** | | | | | | | | | | | | | |
| PRR | 0.0043 *** (9.0651) | | | | | | | | | | | | |
| GVSTAB | | 0.0340 *** (3.3556) | | | | | | | | | | | |
| SOECON | | | 0.0199 *** (9.8632) | | | | | | | | | | |
| INVPROF | | | | 0.0279 *** (6.9518) | | | | | | | | | |
| INTCON | | | | | 0.0520 *** (5.8915) | | | | | | | | |
| EXTCON | | | | | | 0.0604 *** (5.9090) | | | | | | | |
| CORRUP | | | | | | | 0.0208 *** (8.1846) | | | | | | |
| MILPOL | | | | | | | | 0.0253 *** (5.2674) | | | | | |
| RELTEN | | | | | | | | | 0.0178 *** (3.2813) | | | | |
| LAWORD | | | | | | | | | | 0.0012 (0.6759) | | | |
| ETHTEN | | | | | | | | | | | 0.0137 *** (4.4700) | | |
| DEMACC | | | | | | | | | | | | 0.0295 *** (5.9796) | |
| BUREAU | | | | | | | | | | | | | 0.0161 *** (7.0848) |
| POPG | −0.0036 * (−1.8193) | −0.0318 *** (−7.2095) | −0.0085 *** (−5.5069) | −0.0066 *** (−2.9023) | 0.0029 (0.7689) | −0.0124 *** (−5.6276) | −0.0098 *** (−5.5953) | 0.0071 (1.4936) | −0.0054 (−1.5271) | −0.0158 *** (−10.3885) | −0.0109 *** (−5.9343) | 0.0080 * (1.8012) | −0.0031 (−1.3337) |
| EDU | 0.0001 ** (2.5510) | 0.0004 *** (3.0057) | 0.0001 * (1.9266) | 0.0002 *** (3.0288) | 0.0001 * (1.9126) | 0.0001 * (1.6795) | 0.0001 ** (2.3980) | 0.0001 (0.6698) | 0.0002 *** (3.3189) | 0.0002 *** (5.6338) | 0.0002 *** (3.8390) | 0.0002 *** (3.0370) | 0.0001 * (1.9188) |
| TO | −0.0001 (−0.8351) | 0.0005 *** (4.0116) | 0.0001 ** (2.3921) | −0.0001 (−0.6240) | −0.0007 *** (−3.6300) | −0.0003 *** (−3.2409) | 0.0003 *** (4.3214) | −0.0003 ** (−2.4471) | 0.0002 *** (2.6012) | 0.0002 *** (4.6234) | 0.0003 *** (5.6564) | 0.0001 (0.9801) | 0.0002 *** (3.2865) |
| URBN | 0.0006 *** | 0.0010 *** | 0.0005 *** | 0.0007 *** | 0.0011 *** | 0.0019 *** | 0.0004 * | 0.0006 ** | 0.0010 *** | 0.0017 *** | 0.0014 *** | 0.0003 | 0.0006 *** |

**Table A4.** *Cont.*

| | Model 8 | Model 9 | Model 10 | Model 11 | Model 12 | Model 13 | Model 14 | Model 15 | Model 16 | Model 17 | Model 18 | Model 19 | Model 20 |
|---|---|---|---|---|---|---|---|---|---|---|---|---|---|
| **Variables** | | | | | | | | | | | | | |
| | (3.3730) | (5.2787) | (3.3502) | (3.2985) | (5.6526) | (13.8210) | (1.8866) | (2.5285) | (3.6544) | (13.3972) | (9.7789) | (1.0580) | (3.3182) |
| Constant | 0.2632 *** | 0.2325 *** | 0.4368 *** | 0.3012 *** | 0.0758 | −0.0991 | 0.4353 *** | 0.3587 *** | 0.3462 *** | 0.4696 *** | 0.3769 *** | 0.2758 *** | 0.4208 *** |
| | (10.8782) | (2.6637) | (53.9982) | (11.4185) | (1.1137) | (−1.0158) | (44.9678) | (15.2893) | (9.0522) | (49.2509) | (17.2945) | (7.9908) | (40.4199) |
| Observations | 923 | 444 | 923 | 923 | 923 | 1395 | 923 | 937 | 937 | 1395 | 937 | 923 | 937 |
| Anderson canon. Corr. LM statistic | 138.99 (0.000) | 18.33 (0.000) | 167.44 (0.000) | 70.73 (0.000) | 45.86 (0.000) | 55.69 (0.000) | 112.05 (0.000) | 34.08 (0.000) | 28.70 (0.000) | 180.85 (0.000) | 58.02 (0.000) | 45.49 (0.000) | 96.04 (0.000) |
| Cragg –Donald Wald F- statistic | 81.20 [b] | 6.20 [a] | 101.50 [b] | 25.31 [a] | 23.95 [b] | 28.86 [b] | 42.14 [a] | 11.68 [a] | 14.69 [b] | 103.38 [b] | 30.69 [b] | 15.81 [a] | 53.12 [b] |
| Sargan statistic | 0.699 (0.403) | 3.86 (0.135) | 2.90 (0.087) | 2.89 (0.235) | 0.47 (0.494) | 0.01 (0.936) | 5.08 (0.079) | 1.23 (0.541) | 2.98 (0.084) | 0.744 (0.388) | 1.21 (0.272) | 3.80 (0.149) | 3.29 (0.070) |
| Endogeneity test | 35.84 (0.000) | 19.98 (0.000) | 19.13 (0.000) | 62.47 (0.000) | 75.19 (0.000) | 79.66 (0.000) | 44.35 (0.000) | 39.46 (0.000) | 19.12 (0.000) | 8.76 (0.003) | 8.49 (0.004) | 63.76 (0.000) | 17.33 (0.000) |
| Instruments | Latitude EF | Latitude EF SM | Latitude EF | Latitude EF GEL | Latitude EF | Landlocked FO | Latitude EF UKL | EF UKL GEL | EF SCL | SCL UKL | SCL EF | Latitude EF UKL | GRL EF |

Note: All models were estimated using 2SLS estimation. Numbers within parentheses are the t-statistics of coefficients, except in the case of the Anderson canon. corr. LM statistic, Sargan chi-squared, and endogeneity test for which *p*-values are given. [a] Stock–Yogo weak ID test critical values at 10%, 15%, 20%, and 25% of the maximum IV sizes were 22.30, 12.83, 9.54, and 7.80, respectively. [b] Stock–Yogo weak ID test critical values at 10%, 15%, 20%, and 25% of the maximum IV sizes were 19.93, 11.59, 8.75, and 7.25, respectively. ***, **, and * indicate significance at the 1%, 5%, and 10% levels, respectively. EF, SM, UKL, GEL, SCL and FO denote ethnic–fractionalization, Settlers' mortality, English legal origin, German legal origin, Scandinavian legal origin and financial openness, respectively. Source: Author's estimation.

**Table A5.** Impact of institutions on the SDG2 performance according to worldwide governance indicators (robustness check using alternative estimation techniques).

|  | **Model 1** | **Model 1** | **Model 1** | **Model 1** |
|---|---|---|---|---|
| Variables | 2SLS | OLS | FE | RE |
| CGI | 0.0246 *** | 0.0102 *** | 0.0066 ** | 0.0151 *** |
|  | (8.718) | (10.96) | (2.10) | (0.00191) |
| POPG | −0.0012 | −0.0106 *** | −0.0026 | −0.0051 *** |
|  | (−0.561) | (−7.38) | (−1.14) | (0.00190) |
| TO | −0.0000 | 0.0001 *** | −0.0001 * | −0.0001 * |
|  | (−0.124) | (2.91) | (a−1.68) | (−1.69) |
| EDU | 0.0002 *** | 0.0002 *** | 0.00003 | 0.00003 |
|  | (3.494) | (5.09) | (1.22) | (1.20) |
| URBN | 0.0004 ** | 0.0012 *** | −0.0019 *** | 0.0004 ** |
|  | (2.050) | (11.88) | (−4.58) | (1.94) |
| Constant | 0.5496 *** | 0.5053 *** | 0.7215 *** | 0.5684 *** |
|  | (44.159) | (73.25) | (27.39) | (39.09) |
| LM test |  |  |  | 3634.51 |
|  |  |  |  | (0.000) |
| F-test |  |  | 25.07 |  |
|  |  |  | (0.000) |  |
| Hausman test |  |  | 52.62 |  |
|  |  |  | (0.000) |  |
| Observations | 860 | 1325 | 1325 | 1325 |
| R-squared | 0.456 | 0.505 | 0.177 |  |
| Number of codes |  |  | 97 | 97 |

Note: The models were estimated using 2SLS estimation, the OLS method, and fixed effect and random effect models. Numbers within parentheses are the *t*-statistics of coefficients, except for the random effect model for which z-statistics are given. ***, **, and * indicate significance at the 1%, 5%, and 10% levels, respectively. Source: Author's estimation.

**Table A6.** Impact of institutions on the performance of SDG2 according to the political risk rating (robustness check using alternative estimation techniques).

|  | **Model 8** | **Model 8** | **Model 8** | **Model 8** |
|---|---|---|---|---|
| Variables | 2SLS | OLS | FE | RE |
| PRR | 0.0043 *** | 0.0018 *** | 0.0003 | 0.0015 *** |
|  | (9.0651) | (11.39) | (1.29) | (7.24) |
| POPG | −0.0036 * | −0.0115 *** | −0.0025 | −0.0063 *** |
|  | (−1.8193) | (−8.44) | (−1.20) | (−3.28) |
| EDU | 0.0001 ** | 0.0002 *** | 0.00004 | 0.00004 |
|  | (2.5510) | (4.85) | (1.42) | (1.49) |
| TO | −0.0001 | 0.0001 ** | −0.0001 | −0.0001 |
|  | (−0.8351) | (2.31) | (−1.79) | (−1.39) |
| URBN | 0.0006 *** | 0.0012 *** | −0.0019 *** | 0.0008 *** |
|  | (3.3730) | (13.19) | (−4.69) | (3.84) |
| Constant | 0.2632 *** | 0.3843 *** | 0.6992 *** | 0.4450 *** |
|  | (10.8782) | (38.47) | (19.89) | (23.96) |
| LM test |  |  |  | 4230.43 |
|  |  |  |  | (0.000) |
| F-test |  |  | 27.91 |  |
|  |  |  | (0.000) |  |
| Hausman test |  |  | 77.16 |  |
|  |  |  | (0.000) |  |
| Observations | 923 | 1395 | 1395 | 1395 |
| R-squared | 0.457 | 0.507 | 0.312 |  |
| Number of codes |  |  | 97 | 97 |

Note: The models were estimated using 2SLS estimation, OLS method, and fixed effect and random effect models. Numbers within parentheses are the *t*-statistics of coefficients, except the random effect model for which z-statistics are given. ***, **, and * indicate significance at the 1%, 5%, and 10% levels, respectively. Source: Author's estimation.

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
