# Peer review of "Towards the Sustainable Development Goal of Zero Hunger: What Role Do Institutions Play?"

_sustainability, doi:10.3390/su14084598_

Round 1
Reviewer 1 Report
This study, first, revealed the likely impacts that institutions had on SDG2 541 achievement, expanding beyond food security, and hence, second, this research contrib- 542 uted to the empirical literature by examining the precise channels of influence on the 543 overlooked institution–SDG2 nexus, such as direct and indirect effects. Rather than rely- 544 ing on a single measurement index to represent institutions, this research applied world- 545 wide governance indicators and political risk ratings, both as a composite index and as 546 individual indicators to ascertain the overall institutional impact and the relative im- 547 portance of each dimension regarding achieving SDG2. Using simultaneous equation 548 modeling with longitudinal data to yield robust estimates was another distinctive fea- 549 ture of this study.
I have read the revised manuscript for the paper titled “Towards the Sustainable Development Goal of Zero Hunger: What Role Do Institutions Play?”. In my opinion, this is a successful research paper. I would like to applaud the authors for their due diligence and significant effort to answer the questions in their research and improve the quality of their research area.
However, the authors can improve the writing style of the paper by hiring a suitable editor.
Reviewer 2 Report
Esteemed Author,
I am hereby writing my feedback for the scientific paper entitled ”Towards the Sustainable Development Goal of Zero Hunger: What Role Do Institutions Play?.”
The sustainability of food systems is one of the most critical challenges of the 21st century. According to the last data of the Food and Agriculture Organisation (FAO), by the effect of demographic growth and changes in diets and incomes, the demand for food will likely grow by 70% until 2050. The current outlook of the increasingly global market can be affected by considerable uncertainties of supply linked to unpredictable economic and political and climatic, and biological developments. This situation implies a need for accelerated agricultural production growth in developing countries.
The recent reforms of the Common Agricultural Policy (CAP) and other European Union (EU) policies and international and bilateral trade negotiations take into account the objective of global food security. The Joint Research Center (JRC) of the European Commission is involved in the impact assessment of policies regarding food security. Also, the potential trade agreements through economic modeling and the global CGE (Computable General Equilibrium) models assess the economy-wide impacts of the trade policy changes. All these changes are affecting all sectors of the partners. Besides, the global partial equilibrium models simulate the consequences incurred by the agricultural areas of the partners.
In the 2021-27 program for international cooperation, the EU will work on developing the sustainability of food systems with about 70 partner countries. Moreover, at the Nutrition for Growth Summit in Tokyo in December 2021, the EU and its Member States committed to continue addressing malnutrition with a substantial pledge amounting to EUR 4.3 billion, including at least €2.5 billion from the EU for international cooperation with a nutrition objective in the period 2021-2024.
Food security relies on small-scale farming systems in a dual structure in low-income and less industrialized countries. In such situations, the specific modeling at the farm level is essential for estimating the agricultural potential, which is evident in the case of critical areas such as the Black Sea or Sub-Saharan Africa and other regions of the world.
The agricultural production potential in the Euro-Asian/Black Sea cereal-producing region (Russia, Ukraine, etc.) is essential for global food security. This area is studied to estimate the role of this region in grain supply to the world markets and provide insights regarding food security in the short and medium run. Unfortunately, the current conflict situation does nothing but exacerbate existing issues and create new ones.
At the farm level, developing countries are characterized by specific features such as a small scale and a household subsistence character of producers. Therefore particular models need to be prepared to capture in low-income economies and at the farm level, and from there at a regional or national level, the impacts of different policy options.
Agricultural and rural informatization is the basis for the modernization of agriculture; there are many different ways of construction and models in agricultural information systems. At the European Union (EU) level, the European Commission calls on Member states to take advantage of the potential of new technologies and digitization in agriculture. The implementation of new technologies is necessary to improve the sustainability and competitiveness of the sector while simplifying the daily work of farmers. It is essential to achieve the objectives set by the current and future Common Agricultural Policy (CAP) in these new conditions.
At the level of the European Union, to improve food affordability, Member States may also implement reduced rates of Value Added Tax and encourage economic operators to contain retail prices. Member States can also draw from EU funds such as the Fund for European Aid to the Most Deprived (FEAD), which supports EU countries' actions to provide food and essential material assistance to the most vulnerable.
Theory-wise, the paper is likely to elicit the interest of specialists in areas such as economy, conventional food consumption, sociology, sustainable development of agriculture, agricultural productivity, food security, and public policies. The paper presents essential practical applicability primarily related to the economy, sustainable development of agriculture, consumer education, and social awareness of farmers' management in crop production. Moreover, the obtained results are also relevant to the production sector, particularly the industry of certain agri-food products in developing countries.
The paper is well structured and possesses an appreciable novelty character. The major components of the article – Introduction; Literature Review; Methodology; Results and Discussion; Robustness Check and Conclusions - are organized judiciously and directly linked to one another.
The documentation is adequate, and the provided scientific results are precise. The goal of the conducted research is well specified and delineated. The working protocol is appropriate, and the used analysis methods are coherent with the proposed objectives.
The bibliography of the paper is generous. What is even more relevant for the quality of the article, all the authors in the bibliographic reference list are quoted in the text of the material (without exception). However, I suggest that the author revise the list of bibliographic references with the authors in alphabetical order, complete them with the necessary data (where appropriate), and correct minor typos.
Also, taking into account the article's topic, the introduction of brief references on the impact that specific legislation may have on food safety and security may be welcome. In this regard, I suggest that the author consult and include in the list of bibliographic references the following works:
Bondoc, I. European Regulation in the Veterinary Sanitary and Food Safety Area, a Component of the European Policies on the Safety of Food Products and the Protection of Consumer Interests: A 2007 Retrospective. Part Two: Regulations. Universul Juridic, Supliment, 2016, pp. 16-19 (Available online: http://revista.universuljuridic.ro/supliment/european-regulation-veterinary-sanitary-food-safety-area-component-european-policies-safety-food-products-protection-consumer-interests-2007-retrospective-2/).
Bondoc, I. European Regulation in the Veterinary Sanitary and Food Safety Area, a Component of the European Policies on the Safety of Food Products and the Protection of Consumer Interests: A 2007 Retrospective. Part Three: Directives. Universul Juridic, Supliment, 2016, pp. 20-23 (Available online: http://revista.universuljuridic.ro/supliment/european-regulation-veterinary-sanitary-food-safety-area-component-european-policies-safety-food-products-protection-consumer-interests-2007-retrospective-part/).
Bondoc, I. European Regulation in the Veterinary Sanitary and Food Safety Area, a Component of the European Policies on the Safety of Food Products and the Protection of Consumer Interests: A 2007 Retrospective. Part Four: Decisions. Universul Juridic, Supliment, 2016, pp. 24-27 (Available online: http://revista.universuljuridic.ro/supliment/european-regulation-veterinary-sanitary-food-safety-area-component-european-policies-safety-food-products-protection-consumer-interests-2007-retrospective-part-2/).
All these papers approach food safety legislation enforced within the European Union, which usually constitutes a blueprint for the specific law in third countries, especially for the countries with solid trade relations with the European Union. These three documents outline the European legislative environment, starting with 2007, the year of the penultimate geo-political enlargement of the European Union. I want to add that all four recommended papers have been indexed in CAB International and HeinOnline, the most extensive worldwide database for documents in the legal field.
All three articles represent a systematic database regarding all the normative acts issued and applicable at the European Union level in 2007 in food security, food safety, nutrition (including additives and food supplements), and public health.
The work also benefits from adequate iconographic support, materialized by five tables and six figures (with the possibility of introducing new tables).
The author should pay more attention to certain abbreviations to avoid confusion; basically, all abbreviations are to be used in the text-only after at least one mention made in extenso.
The obtained results are interpreted correctly, and their practical value is visible.
The graphical representation of the results is adequate; as for the grammar of the paper, most of the text is very well written, with very few parts that would require some minor corrections, as follows:
Page 3, line 120 – replace “to creating” with “to create”;
Page 4, line 166 – replace “negatively impacts” with “negatively impact”;
Page 19, line 541 – add an extra line between “Conclusion” and “This study” in “ConclusionThis study”.
Minor corrections and clarifications notwithstanding, the author’s work and results are highly commendable. They add significant value to the paper and may constitute a launching pad for further valuable studies.
Provided that the author verifies the paper and performs the required corrections, the article can be accepted and published in the Sustainability.
Best Regards,
Reviewer
Author Response
Distinguished reviewer
Please see the attachment.
